# Impact of Environmental Conditions on the Degree of Efficiency and Operating Range of PV-Powered Electric Vehicles

**Christian Schuss** * and **Tapio Fabritius**

Optoelectronics and Measurement Techniques (OPEM) Research Unit, University of Oulu,
FIN-90570 Oulu, Finland; tapio.fabritius@oulu.fi
* Correspondence: christian.schuss@oulu.fi; Tel.: +358-294-482754

**Abstract:** This paper investigates the impact of environmental conditions on the possible output power of photovoltaic (PV) installations on top of hybrid electric vehicles (HEVs) and battery-powered electric vehicles (BEVs). First, we discuss the characteristics and behavior of PV cells in order to provide an understanding of the energy source that we aim to integrate into vehicles. Second, we elaborate on how PV cells and panels can be simulated to estimate the potential extension of the electrical driving range (ERE) of BEVs and HEVs. In particular, we concentrate on the impact of the vehicle's curved roof surface on the possible output of the PV installation. In this research, we present considerations for vehicles in both parking and driving conditions. More precisely, we demonstrate how the frequently changing environmental conditions that occur while driving represent significant challenges to the control of the operating voltage of PV cells. As the area for deploying PV cells on top of an electric vehicle is limited, attention needs to be paid to how to optimize and maximize the degree of efficiency of PV-powered electric vehicles.

**Keywords:** battery-powered electric vehicle (BEV); charging; data acquisition; environmental data; hybrid electric vehicle (HEV); measurement; photovoltaic (PV) cell; photovoltaic (PV) energy; simulation

## 1. Introduction

Due to the electrification of transportation for human society, it is forecasted that the transportation sector will account for a significant share of society's energy demand in the future. Recent statistics clearly indicate a rapid increase of plug-in hybrid electric vehicles (HEVs) and battery-powered electric vehicles (BEVs). In the future, connected and autonomous vehicles (CAVs) will also need to be charged by the electricity grid; thus, they will increase the loads and challenges for smart grids [1–3].

When conceptualizing future transportation, the energy demand of the type of transportation needs to be taken into account. Forecasts indicate that society's energy consumption will increase by 56% between 2010 and 2040 [4]. Even though new propulsion and driving technologies are being implemented in new automobiles, for charging plug-in HEVs, BEVs and CAVs, the vast majority is produced by non-renewable resources, such as coal, oil, gas and nuclear power (COGN) energy sources [5,6].

Overall, estimations show that global energy-related carbon dioxide emissions by human society will increase to 45 billion metric tons by the year 2040, representing an increase of 46% compared to 2010 [4]. Solar energy has been identified as one of the most promising candidates to help withdraw from the dependency on COGN energy sources [6], and it can be used for large-scale and small-scale energy production [7,8].

Hence, solar energy provides the opportunity to produce the required electricity for plug-in HEVs, BEVs and CAVs. More precisely, photovoltaic (PV) cells can be integrated into the roof and other surfaces of electric vehicles in order to provide electricity for the onboard power supply and to extend the electrical driving range (ERE) [9,10]. Likewise, PV charging stations can be implemented for charging the high-voltage batteries of BEVs, for example [7,11,12].

When integrating PV cells into the surfaces of electric vehicles, these types of vehicles can be turned from consumers of electricity from the power grid into producers of energy for the grid [9]. There is an opportunity to produce PV energy under parking conditions, as well as driving conditions [13–15]. In this way, the contribution of electric vehicles to the pollution of the environment can also be reduced [16].

Statistics show that in 2015, 950 million vehicles for individual transportation were in operation. Statistics also show that the number of passenger vehicles is increasing at a steady rate every year [17]. Hence, the research and development (R&D) questions that arise when trying to integrate PV cells into electric vehicles in an efficient way must be addressed [16].

At present, PV energy production in electric vehicles represents a new area for this type of resource [10,16]. Hence, as with other new application fields for energy resources, the given environmental conditions have to be studied and investigated. More precisely, it is crucial to understand the behavior and fluctuations of the available output current and power of PV cells due to changes in the available irradiation [18].

Furthermore, the potential contribution of vehicles' PV installations to the overall electricity demand can be calculated. Here, PV simulation models can help to predict the possible extension of the ERE of electric vehicles. Recent research indicates that, depending on the size of the PV installation and the environmental conditions, about 70% of a vehicle's electricity demand can be met with PV energy [16].

The single-diode model is used as an equivalent circuit to model single PV cells and panels [8,19–21]. The accuracy of the PV simulation model depends on two tasks. First, unknown model parameters have to be calculated and estimated. In this way, the error between real and simulated PV cells should be as small as possible [19–23].

Second, it has to be taken into account that the environmental conditions have a direct impact on the obtainable output current, and thus the potential output power of photovoltaics. Hence, it is advisable to include environmental data at a suitable resolution in order to simulate the output power of a PV cell. If environmental data at a lower sample rate are obtained, the data could lead to false estimates [24].

Generally speaking, the available area and surface on top of an electric vehicle for the deployment and integration of PV cells is limited [18]. Areas other than the roof of the vehicle, such as the side, the rear and the front, receive considerably less irradiation [25]. In addition, the circumstances, particularly the rate of change of irradiation, differ significantly between driving in urban areas and leaving the vehicle parked [18,25].

As part of its research work, Toyota recently presented a solar-powered version of the Prius. Integrated into the vehicle's hood, roof and rear hatch were 1188 PV cells manufactured by Sharp, achieving a conversion efficiency of more than 34%. The number of PV cells could be increased further if other areas of the Prius were also used [26]. However, as the Prius is a plug-in HEV, the capability of storing the obtained PV energy (e.g., in the vehicle's high-voltage battery) can be limited [27–29].

In this paper, we summarize and categorize our previous research and other related research investigating the environmental impact on the degree of efficiency of PV installations integrated in electric vehicles today and in the future. The focus is on potential ways to maximize and optimize the ERE of plug-in HEVs, BEVs and CAVs. Hence, we present tools and methods that allow us to address the given circumstances (e.g., curved surfaces, parking and driving conditions) and ambient conditions involved in PV installation in electric vehicles.

In this paper, the goal is to define the main impacts on the degree of efficiency of PV-powered electric vehicles and to emphasize the importance of understanding environmental conditions that affect PV cells output behavior and performance. Hence, the main contributions of this work are as follows:

(1) A PV simulation model is proposed which allows us to take into consideration the different alignment of PV cells on top of curved surfaces, such as the vehicle's roof, and the vehicle's cardinal orientation towards the sun. In contrast to other models in

the available literature, the proposed model in this work allows us to estimate and predict accurately the potential energy of PV-powered electric vehicle.

(2)  The impact of environmental conditions on PV-powered electric vehicles was studied. While in parking conditions, the circumstances are comparable to PV-powered charging stations, in driving conditions, the available irradiation is changing quickly and rapidly. Example data are presented for PV-powered vehicles in both parking and driving conditions.

(3)  The challenges to optimize and maximize the degree of efficiency of PV-powered electric vehicles are elaborated. On the example of maximum power point tracking (MPPT) in driving conditions, in comparison to vehicles in parking conditions, the potential reduction in system efficiency is analyzed. A possible solution is proposed to control the operating voltage of PV cells on top of curved surfaces in an effective way.

This paper is organized as follows: Section 2 describes the characteristic output behavior of a PV cell. Section 3 presents a suitable simulation model for PV installation embedded in curved surface shapes. Section 4 presents the environmental conditions for PV-powered vehicles in both parking and driving conditions. Section 5 analyzes and discusses the potential degree of efficiency of photovoltaics integrated in electric vehicles. Finally, the conclusion and future works are given in Section 6.

## 2. Characteristic Output Behavior of a PV Cell

The non-linear current–voltage (*I–V*) curve illustrates the characteristic behavior of the output current of a PV cell. The output current varies when the output voltage level is altered [30,31], as illustrated in Figure 1 for a standard 156 mm × 156 mm poly-Si PV cell. Likewise, the power–voltage (*P–V*) curve describes the possible output power ($P_{out}$) at different operating voltages ($V_{op}$).

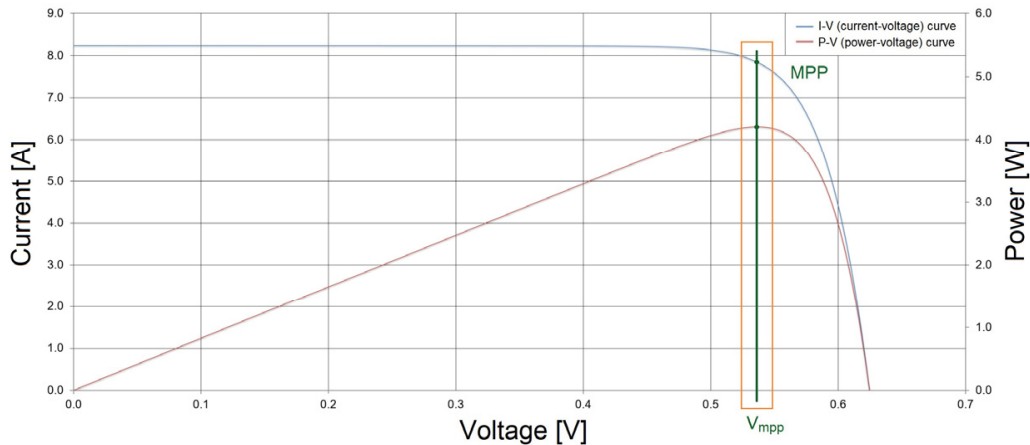

**Figure 1.** *I–V* and *P–V* of a PV cell at STC.

On the *I–V* curve, one point exists at which the product of current and voltage becomes a maximum. This point is commonly referred to as the maximum power point (MPP). The MPP can also be found on the *P–V* curve, as seen in Figure 1. The power of an ideal PV cell or panel ($P_{ideal}$) can be derived from multiplying the open-circuit voltage ($V_{oc}$) and the short-circuit current ($I_{sc}$). The maximum power of a real PV cell, also referred to as the power at the MPP ($P_{mpp}$), can be derived by multiplying the current ($I_{mpp}$) and voltage ($V_{mpp}$) at the MPP.

The available power at the MPP ($P_{mpp}$) greatly depends on the solar radiation level ($\lambda$) and PV cell temperature ($T_c$) [18,20–23,30,31]. At standard test conditions, with $\lambda = 1000 \, W/m^2$, $T_c = 25 \, °C$ and AM1.5, the PV cell produces about 4 watts. However, at lower solar radiation

levels, the power is reduced quickly. For example, with $\lambda$ = 200 W/m$^2$, less than 1 watt can be obtained.

When driving in an urban area, shading from the surrounding environment can represent challenges for the MPPT algorithm [10,13]. Generally speaking, the MPPT algorithm modifies $V_{op}$ on a continuous basis in order to stay as close as possible to $V_{mpp}$; hence, it aims to maximize the possible power and degree of efficiency of PV energy production [32]. For some MPPT techniques, the quickly and rapidly changing environmental conditions during driving can result in a significant reduction of obtained output power [13].

## 3. Computer Simulation of PV Cell and Panel

### 3.1. Purpose of Computer Simulations

As seen in Figure 1, the *I–V* and *P–V* curves show the potential output current and power of a PV cell under STC ($\lambda$ = 1000 W/m$^2$, $T_c$ = 25 °C and AM1.5). However, as the environmental conditions outdoors vary, so do the output current and power of PV cells. Hence, simulations are helpful in calculating the output performance under different ambient conditions than the STC.

With the help of computer simulations, commonly carried out in MATLAB/Simulink or other mathematical programs, we are able to obtain information on how much PV energy we can produce with PV cells that are integrated into the surfaces of electric vehicles. Moreover, simulations can provide estimates on how much we can charge high-voltage batteries in parking conditions and prolong the ERE in driving conditions.

As seen in Figure 1, a single PV cell provides only a certain level of output voltage ($V_{out}$) and output current ($I_{out}$). Hence, two PV cells are connected either in series to double the $V_{out}$ or in parallel to double the $I_{out}$. In this way, PV cells are connected with each other to form a PV panel. Commonly, in PV simulation models, series and parallel connections are considered by the factor $N_s$ (number of PV cells in series) and $N_p$ (number of PV cells in parallel). For a single PV cell, Ns and Np are equal to 1. Different interconnections do not change the shape of the *I–V* curve, but the scale of the *x*-axis (series connection) or *y*-axis (parallel connection).

However, this assumption can only be made for a flat PV panel in which all PV cells, despite their type of interconnection, are aligned under the same angle towards the sun. Figures 2 and 3 show the curved shape of the roof of a Toyota Prius. The length of the roof area is about 168 cm, and the width is about 108 cm. If standard 156 mm × 156 mm silicon-based PV cells are used, about 45 cells (nine rows with five cells in each row) can be integrated on the roof.

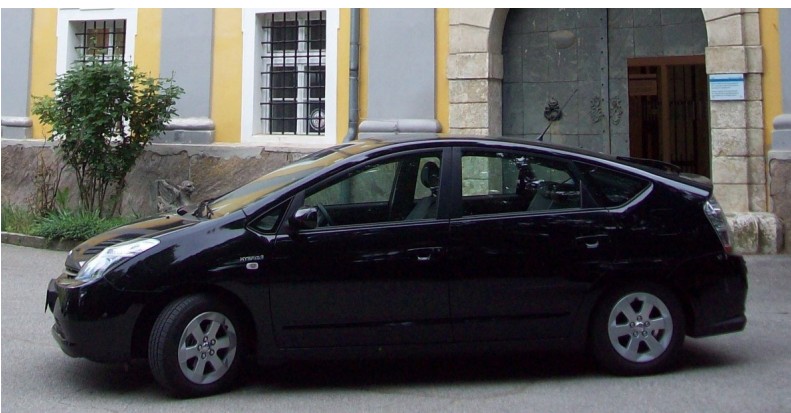

**Figure 2.** Curved shape of the roof of a Toyota Prius.

It is worth noting that the rows of PV cells, from the hood to the rear of the vehicle, will be oriented towards the sun under different angles. Table 1 summarizes the longitudinal angles for rows 1–9. If the vehicle is parked in such a way that the windscreen faces the sun, then the PV cells in rows 1–5 will be oriented towards the sun, while in the rear of the

vehicle, the cells in rows 7–9 will face away from the sun. If the vehicle is turned around 180°, then the rear of the vehicle will be oriented towards the sun, resulting in the opposite situation.

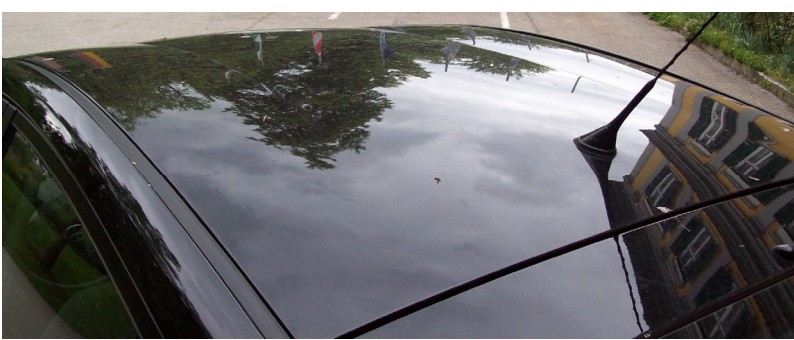

**Figure 3.** Closer look at uneven curved shape of Toyota Prius roof.

**Table 1.** Longitudinal angles of PV cells on the roof of a Toyota Prius.

|       | Row (1) | Row (2) | Row (3) | Row (4) | Row (5) | Row (6) | Row (7) | Row (8) | Row (9) |
|-------|---------|---------|---------|---------|---------|---------|---------|---------|---------|
| angle | 15°     | 12°     | 9°      | 3°      | 2°      | 0°      | −3°     | −7°     | −9°     |

### 3.2. Review of PV Simulation Models

### 3.2.1. Parameters for the Models

The single-diode model is commonly used for simulating silicon-based PV cells [8,19–21]. In the available literature, depending on the information of the PV cells, different types of models can be found. Basically, we can obtain the required parameters, such as the open-circuit voltage ($V_{oc}$) and the short-circuit current ($I_{sc}$), from the datasheet of the photovoltaic manufacturer. Additionally, we can assume certain parameters, such as the ideality factor (A), with typical values to establish simulations. In order to improve the accuracy, different parameter identification techniques can be used to estimate more suitable values for parameters [20].

### 3.2.2. Ideal Model

If only a few parameters from PV cells are available, then the ideal model can be used. However, the accuracy can be low compared to datasets (which illustrate the output voltage and current under different solar radiation and temperature levels). In the ideal model, we assume the PV cell to be equal to a current source, which creates a photocurrent ($I_{ph}$) in direct proportion to solar radiation, and a diode, as shown in Figure 4. The output current ($I_{out}$) is restricted by the diode current ($I_d$). We obtain the output current ($I_{out}$) as follows:

$$I_{out} = N_p\, I_{ph} - N_p\, I_d \tag{1}$$

which, with the Shockley diode equation, is as follows:

$$I_{out} \ = \ N_p\, I_{ph} - N_p\, I_s(e^{\frac{qV_{out}}{N_s\, A k T_c}} - 1) \tag{2}$$

where $I_s$ is the saturation current of the PV cell, $q$ is the charge of an electron and $k$ is the Boltzmann constant. The photocurrent ($I_{ph}$) is calculated by using the following equation:

$$I_{ph} \ = \ I_{sc,ref}\,(1 + K_I(T_c - T_{ref}))\frac{\lambda}{\lambda_{ref}} \tag{3}$$

where $I_{sc,ref}$ is the short-circuit current of the PV cell under reference conditions, $K_I$ is the temperature coefficient for the current, $T_{ref}$ is the reference temperature of PV the cell and

$\lambda_{ref}$ is solar radiation under reference conditions. The saturation current ($I_s$) of the cell is obtained by the following equation:

$$I_s = I_{rs} \left( \frac{T_c}{T_{ref}} \right)^3 e^{qE_g \left( \frac{1}{T_{ref}} - \frac{1}{T_c} \right)} \qquad (4)$$

where $I_{rs}$ is the reverse saturation current of the PV cell, and $E_g$ is the band gap energy of the semiconductor used in the cell in electron volts. The reverse saturation current ($I_{rs}$) is calculated by the following equation:

$$I_{rs} = \frac{I_{sc}}{e^{\frac{qV_{oc,ref} \, (1 + K_V(T_c - T_{ref}))}{N_S A k T_c}} - 1} \qquad (5)$$

where $V_{oc,ref}$ is the open-circuit voltage of the PV cell under reference conditions, and $K_V$ is the temperature coefficient for the voltage.

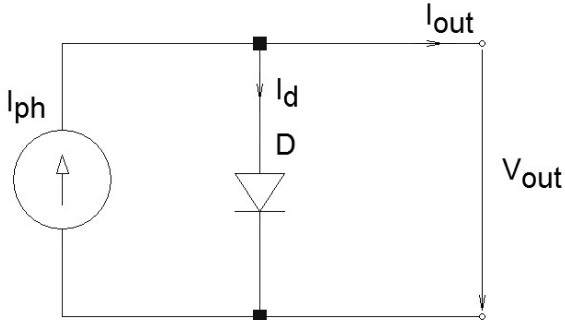

**Figure 4.** Equivalent circuit of ideal PV simulation model.

### 3.2.3. Simplified Model

In order to obtain higher accuracy, series resistance ($R_s$) can be considered, which represents the internal resistance, or, more precisely, the current path through the semiconductor material, the contacts, the metal grid and the current collecting bus. This magnitude of this parameter depends on the type of application in which the PV cells are used. The equivalent circuit is shown in Figure 5. The loss of $R_s$ is taken into consideration in Equation (2) as follows:

$$I_{out} = N_p I_{ph} - N_p I_s \left( e^{\frac{q(N_p V_{out} + N_S I_{out} R_s)}{N_p N_S A k T_c}} - 1 \right) \qquad (6)$$

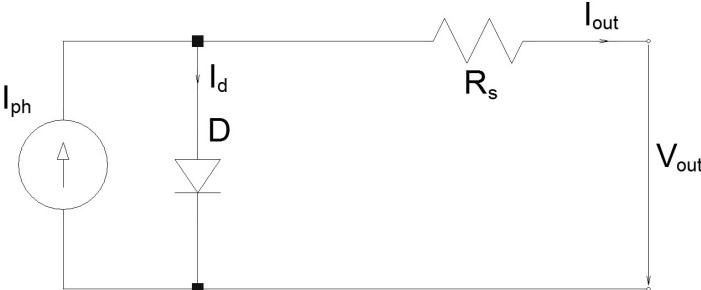

**Figure 5.** Equivalent circuit of simplified PV simulation model.

### 3.2.4. Advanced or Practical Model

For silicon-based PV cells, the advanced or practical model offers a reasonable degree of accuracy for simulations. It considers shunt resistance ($R_{sh}$), which represents the leakage

current of PV cells. The equivalent circuit of the advanced or practical model is shown in Figure 6. An influence of $R_{sh}$ is considered in Equation (6) with the following extension:

$$I_{out} = N_p I_{ph} - N_p I_s (e^{\frac{q(N_p V_{out} + N_s I_{out} R_s)}{N_p N_s A k T_C}} - 1) - \frac{\frac{N_p}{N_s} V_{out} + I_{out} R_S}{R_{sh}} \qquad (7)$$

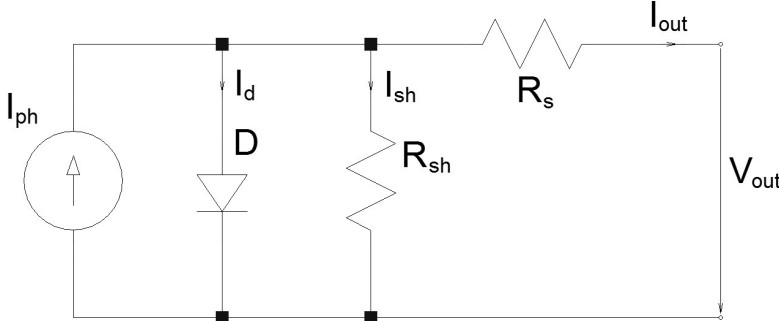

**Figure 6.** Equivalent circuit of advanced or practical PV simulation model.

As mentioned above, in order to simulate an entire PV panel, we can obtain parameters, such as $V_{oc,ref}$, $I_{sc,ref}$, $T_{ref}$, $\lambda_{ref}$, $K_V$, $K_I$, $N_s$ and $N_p$, from the data sheets and specifications of PV manufacturers. However, due to the curved shape of the roof, it is not advisable to use the factors $N_s$ and $N_p$ for simulating PV cells integrated into curved surfaces. Instead, it is advisable to simulate each PV cell individually with the given environmental conditions for each cell [33].

### 3.2.5. Used Simulation Model in this Research

In this work, we used the advanced or practical model to obtain the best possible accuracy within simulations. The model was extended, as described in Section 4, to take into consideration different alignments of PV cells towards the sun and the cardinal direction of the vehicle. Furthermore, the model was used to calculate the $V_{mpp}$, $I_{mpp}$ and $P_{mpp}$ under different solar radiation levels and $V_{oc}$, $V_{out}$ and $I_{out}$ within MPPT simulations in Section 5.

## 4. Environmental Conditions for PV-Power Electric Vehicles
### 4.1. Parking Conditions

If electric vehicles with integrated photovoltaics are parked in urban areas, then the conditions are somewhat comparable to PV-powered charging stations. However, a charging station powered by PV panels will have the panels aligned with the sun to maximize the possible PV energy. At a public parking lot, as seen in Figure 7, drivers often do not have the freedom to choose between available parking spaces. As such, PV-powered electric vehicles will be oriented in different cardinal directions relative to the sun (e.g., in Figure 7, the gray vehicle is oriented to the northeast, the red vehicle to the southwest, etc.), which is not ideal for PV energy production.

As mentioned above, due to the curved shape of the vehicle's roof, the majority of PV cells on top of the roof may be facing away from the sun; thus, the potential output power from the PV cells is reduced. More precisely, the vehicle's orientation relative to the sun affects the effective area ($A_{eff}$) of the PV cells [33,34]. $A_{eff}$ considers the solar azimuth angle ($\theta$) and the cardinal direction ($\psi$), calculated as follows:

$$A_{sff} = cos(\alpha)sin(\beta)cos(\psi - \theta) + sin(\alpha)cos(\beta) \qquad (8)$$

where $\alpha$ is the angle of the solar altitude, and $\beta$ is the longitudinal angle of the PV cell (as seen in the example of a Toyota Prius in Table 1). For $\psi = 0°$, the vehicle's front is oriented towards the north, while for $\psi = 180°$, it is oriented towards the south. Data on the solar radiation level ($\lambda$), which is obtained at a horizontal level ($\beta = 0°$) [24], can be multiplied

with $A_{eff}$ to obtain the solar radiation level at different longitudinal angles. Likewise, the impact on $T_c$ of different longitudinal angles ($\beta \neq 0°$) can be taken into account [34].

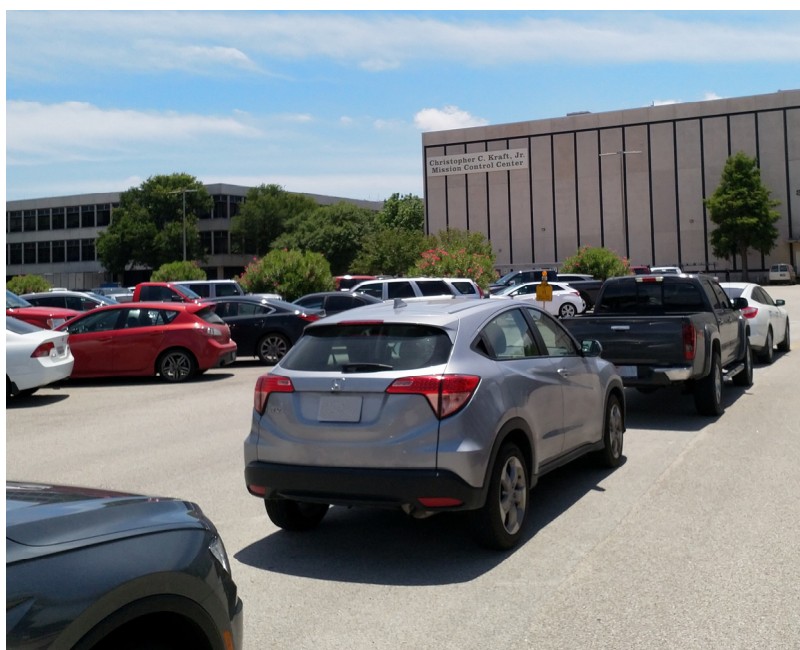

**Figure 7.** Example of parking conditions in a public parking lot.

Figures 8–10 illustrate $A_{eff}$ for $\psi = 180°$ (= the front of the vehicle is oriented towards the south), $\psi = 135°$ (= the front of the vehicle faces southeast) and $\psi = 0°$ (= the front of the vehicle is aligned towards the north). As seen in Figures 8–10, the position of the vehicle has an impact on $A_{eff}$ and, hence, the potential amount of power that can be obtained from a PV cell in a particular row.

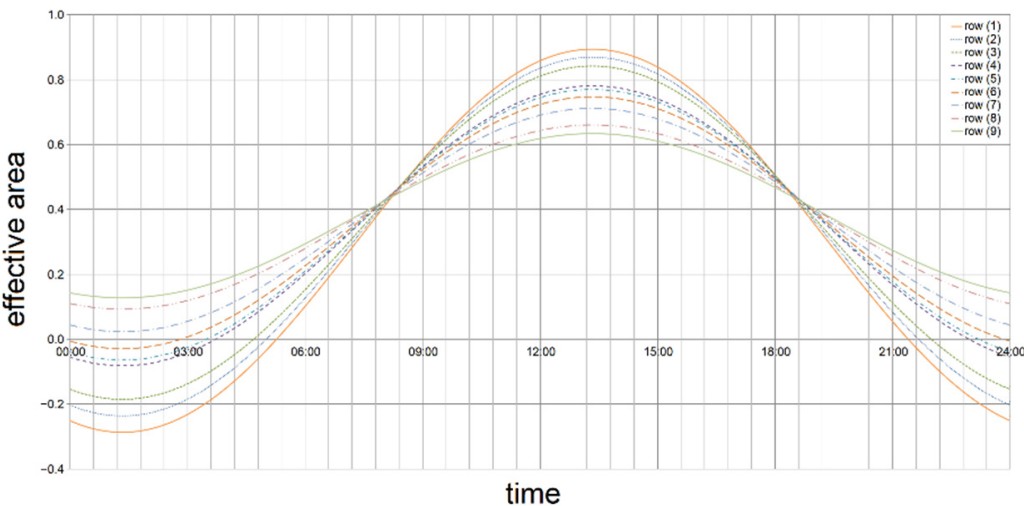

**Figure 8.** Effective area for $\psi = 180°$.

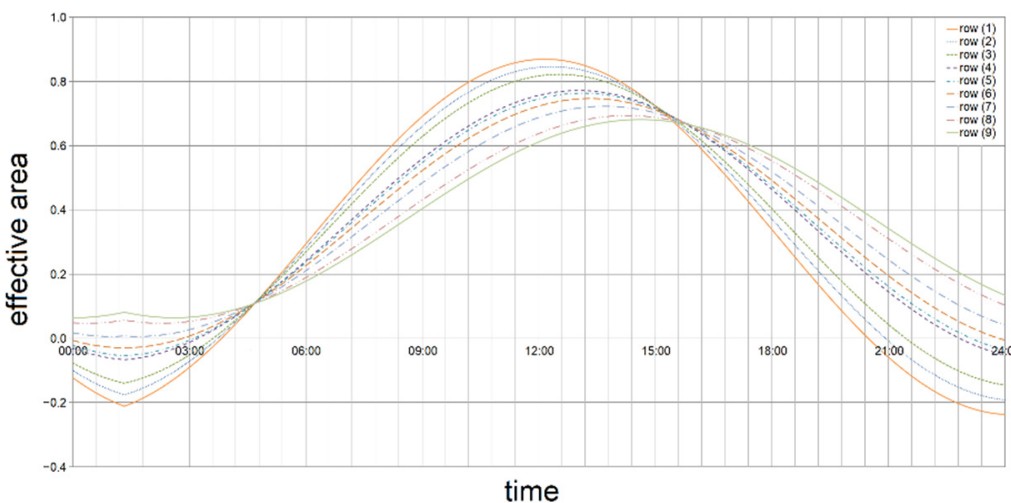

**Figure 9.** Effective area for $\psi = 135°$.

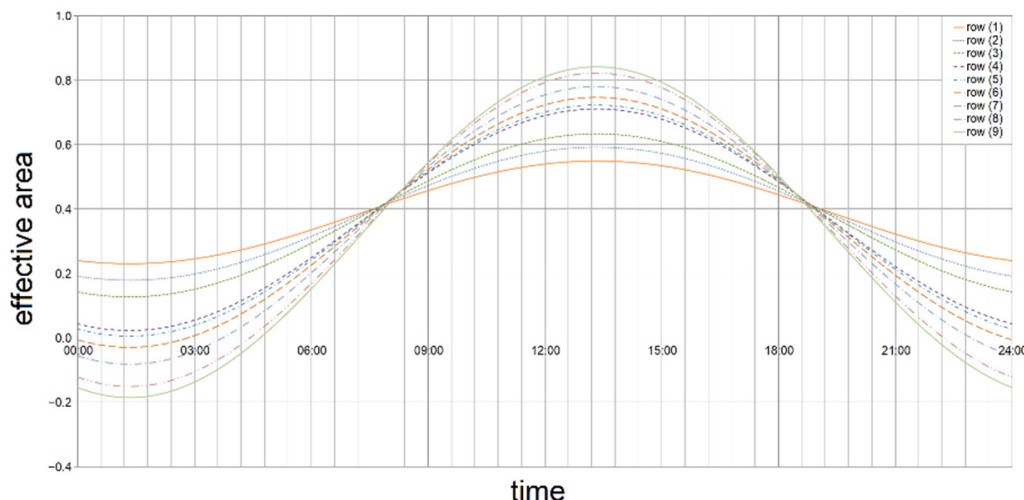

**Figure 10.** Effective area for $\psi = 0°$.

*4.2. Examples of Environmental Data in Parking Conditions*

Figure 11 presents the available solar radiation data ($\lambda$) and PV cell temperature ($T_c$) for 16 June in Oulu, Finland. As seen in this figure, the day was mostly sunny. Figure 12 shows the available solar-radiation data ($\lambda$) and PV cell temperature ($T_c$) for 19 June in Oulu. As seen in this Figure 12, the day was mostly cloudy.

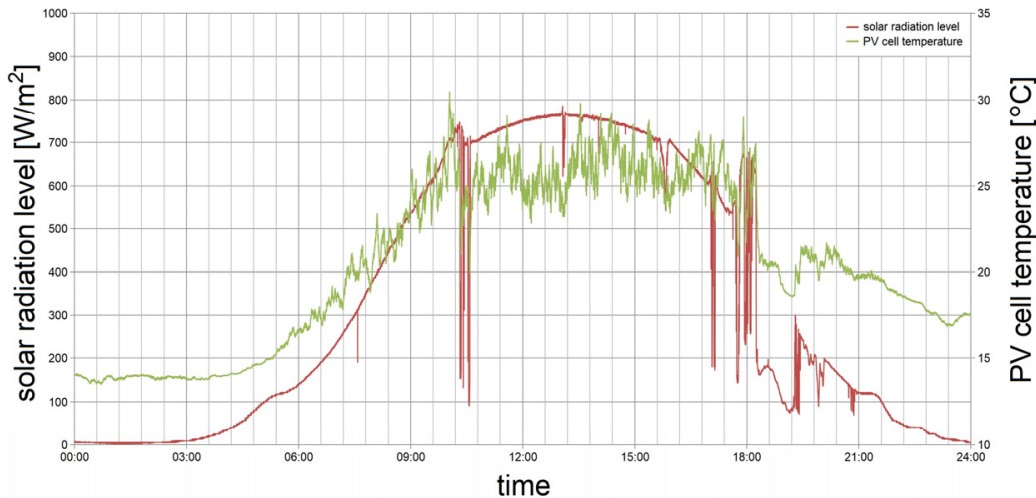

**Figure 11.** $\lambda$ and $T_c$ for 16 June.

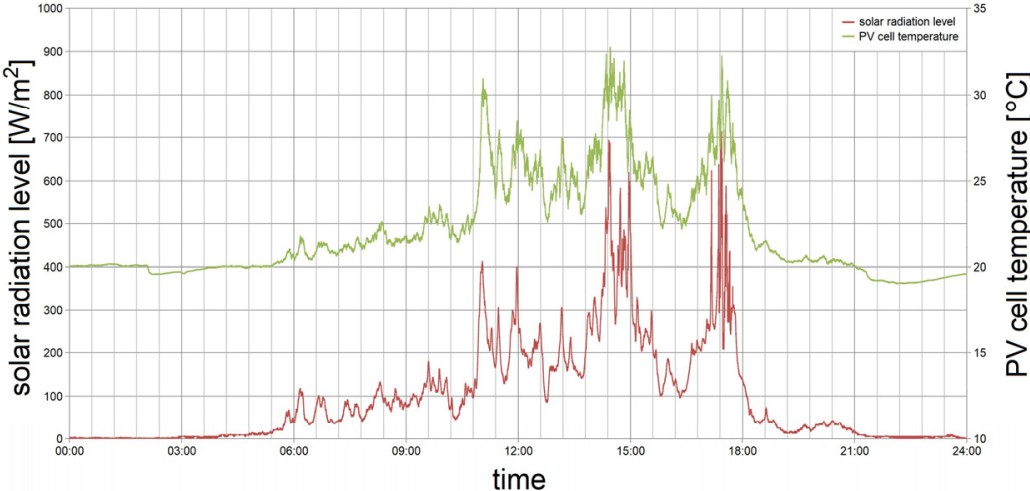

**Figure 12.** $\lambda$ and $T_c$ for 19 June.

### 4.3. Driving Conditions

While the parking of PV-powered electric vehicles represents stationary conditions for PV installation, the driving represents moving or dynamic conditions [13,15]. When driving in an urban area, there will be less PV energy gained than when the car is parked. The environment around the street (e.g., trees, buildings, etc.) will produce shade on top of the PV installation of the vehicle, resulting in lower solar radiation ($\lambda$).

Shade can vary depending on the geographical location, time of the year, time of the day, wind speed and other factors. Figures 13–16 show some examples of shady conditions on Yliopistokatu Street in an urban area of the city of Oulu, Finland. Figures 13 and 15 were taken in spring at 2:00 p.m. local time, and Figures 14 and 16 were taken in summer at 4:00 p.m. It can be seen that there is generally much more shade from trees on the street in summer than in spring.

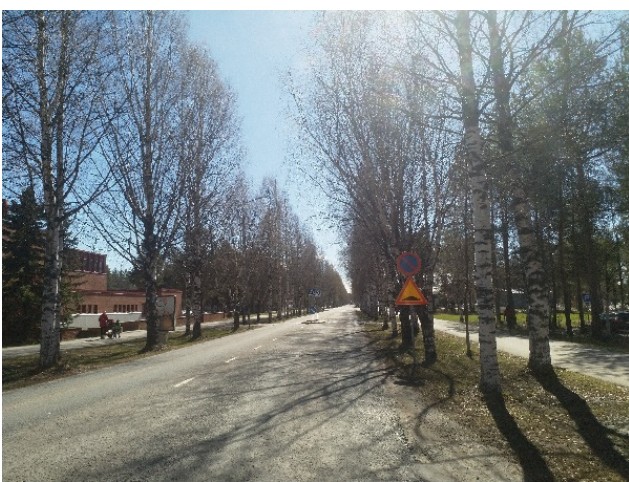

**Figure 13.** Conditions in spring (location 1).

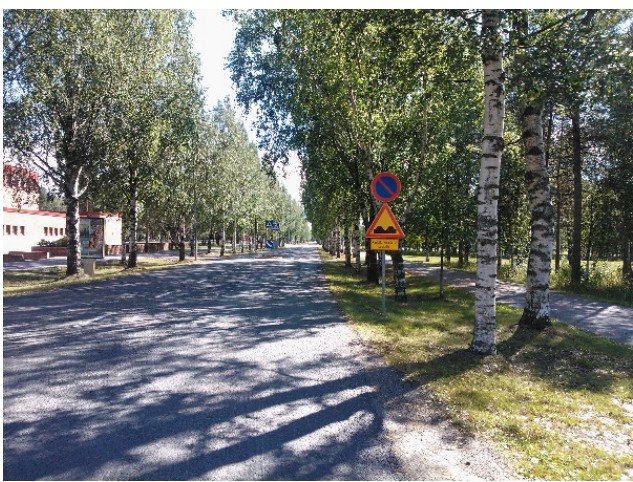

**Figure 14.** Conditions in summer (location 1).

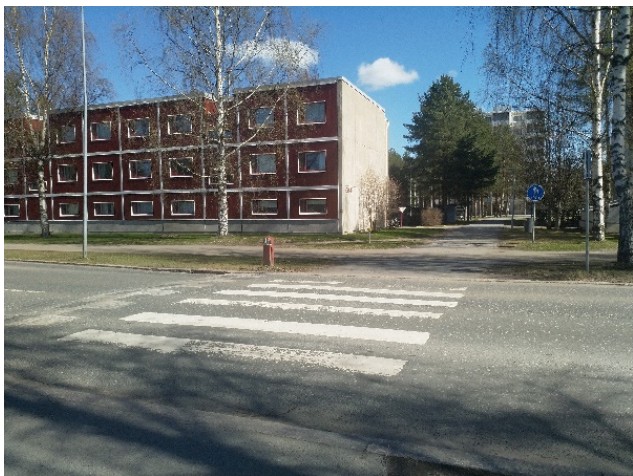

**Figure 15.** Conditions in spring (location 2).

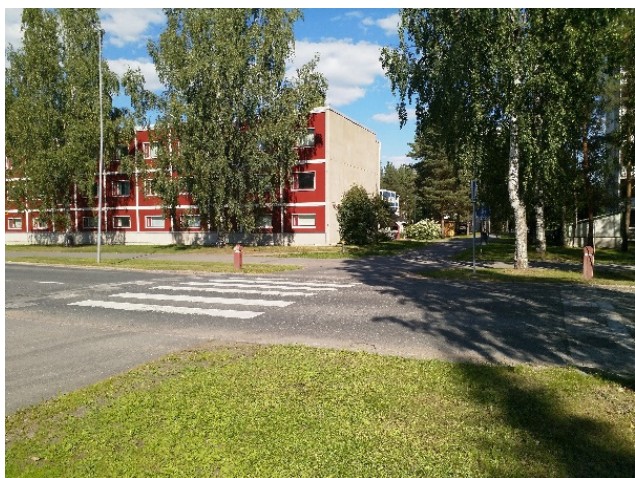

**Figure 16.** Conditions in summer (location 2).

Shade from surrounding objects can occur with a significant contribution of diffuse sunlight (e.g., trees) or instantaneously without such a contribution (e.g., buildings and other constructions). Figure 17 shows Tietolinja Street in Oulu as an example of shade from a highway bridge on which the lanes are separated from each other, resulting in a small area of sunlight between the two lanes [15]. Figure 18 shows the ambient conditions on Victoria Street in Auckland, New Zealand. Shade can also be caused normally from clouds. Here, even the shape of the street causes additional differences in the alignment of PV cells towards the sun.

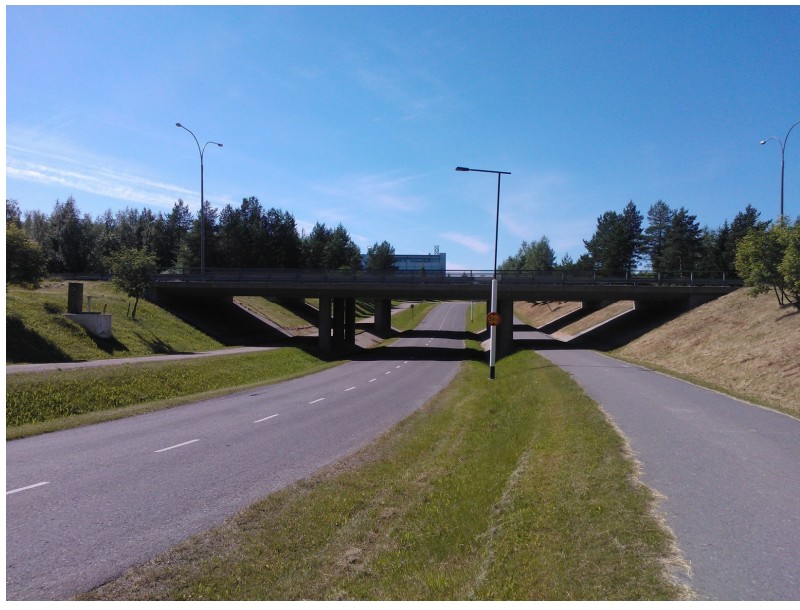

**Figure 17.** Example of shade from a highway bridge.

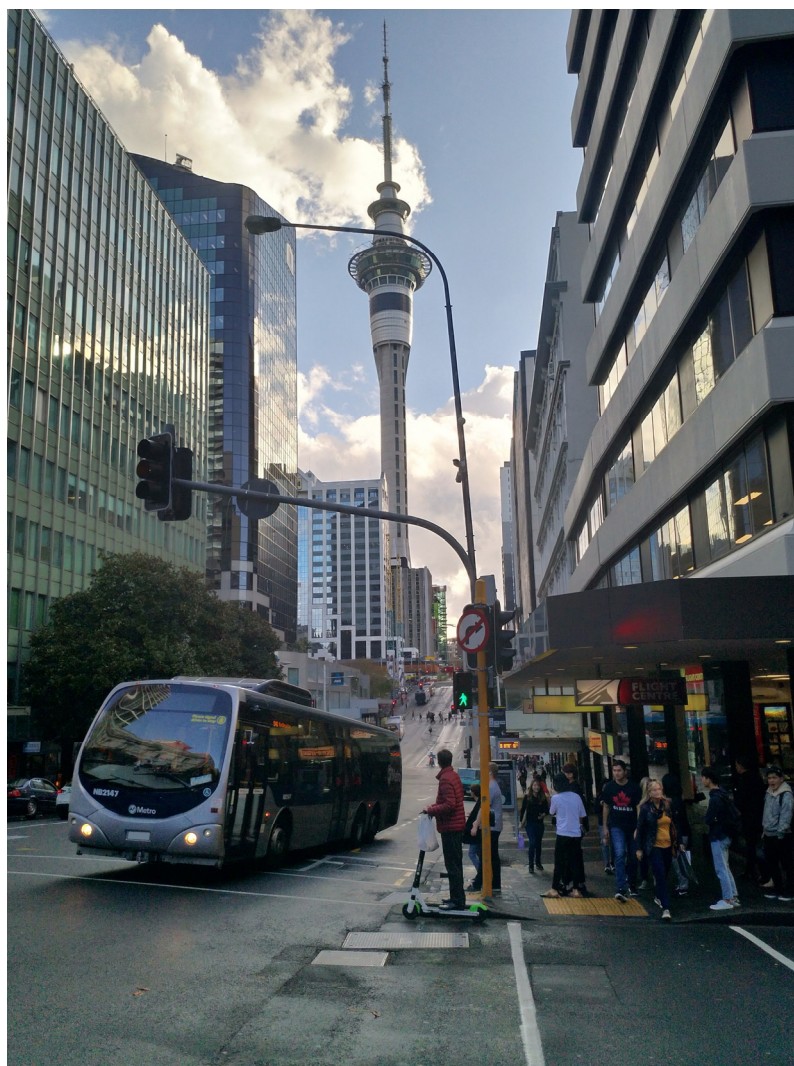

**Figure 18.** Example of conditions in other geographical areas.

*4.4. Examples of Environmental Data in Driving Conditions*

Figures 19 and 20 present data obtained under driving conditions. Similar to Reference [18], the vehicle with the measurement setup was driving from the sports hall located south of the University of Oulu to the botanical garden located north of the university on Yliopistokatu Street. Figure 19 shows the data obtained when driving from the sports hall to the botanical garden, and Figure 20 shows the data obtained on the way back. Even though the vehicle was driving the same route, the driving direction had a significant impact on the obtained solar radiation level.

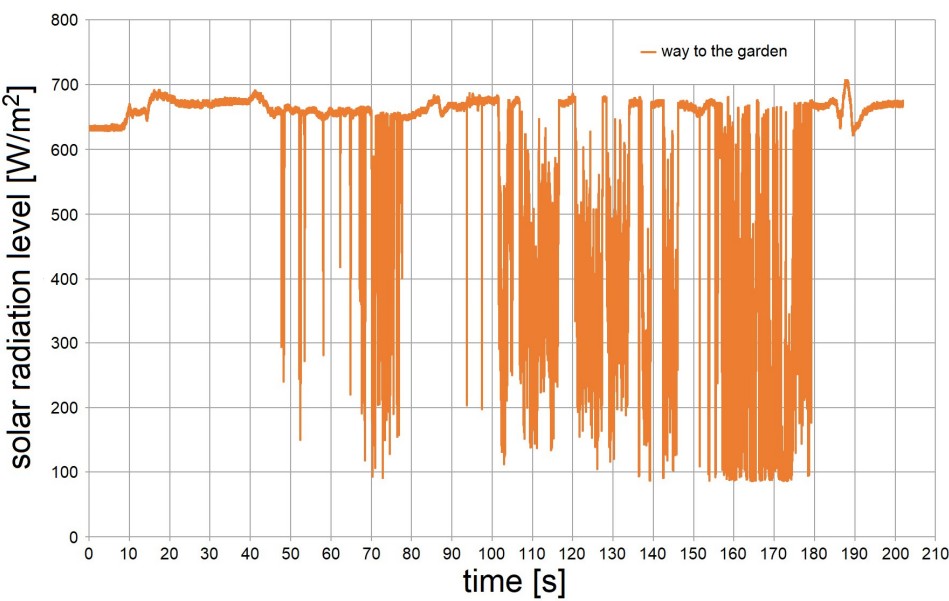

**Figure 19.** $\lambda$ for trip from sports hall to botanical garden.

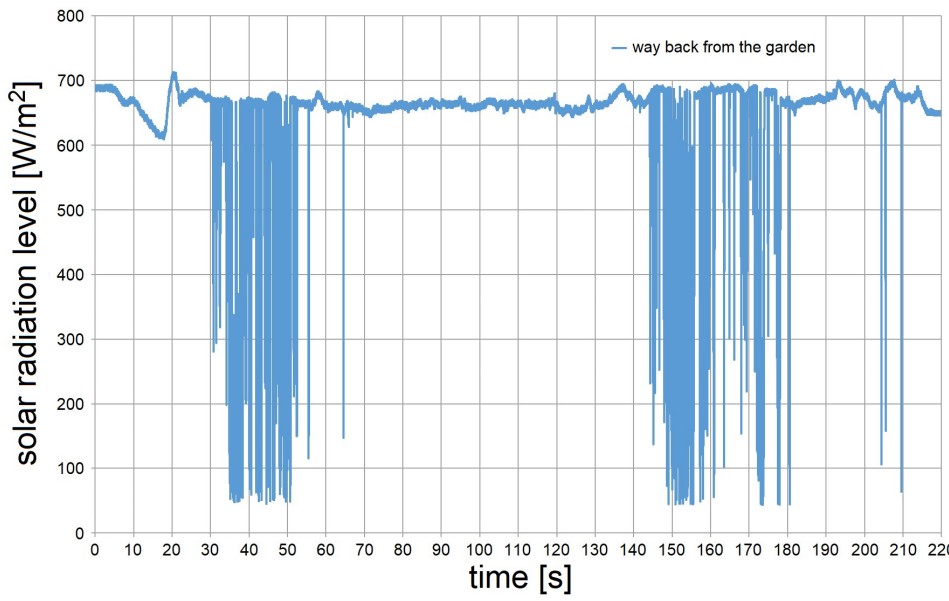

**Figure 20.** $\lambda$ for trip from botanical garden to sports hall.

## 5. Analyses and Discussion

### 5.1. Analyses of the Impacts on the Degree of Efficiency of PV-Powered Vehicles

If generally better environmental conditions can be obtained at PV-powered charging stations as compared to PV-powered vehicles, as proposed in References [7,11,12], the question as to why we should focus on PV installations in electric vehicles arises. First, if a PV-powered charging station is only available at the vehicle owner's home, then the car cannot be charged, for example, when the owner takes the car to work. As such, the PV energy from the PV-powered charging station needs to be either supplied to the power grid or stored in another battery until the vehicle's high-voltage battery can be charged.

As seen in Figure 7, when choosing a parking space, the driver has an influence on the cardinal direction of the vehicle ($\psi$) relative to the sun. Commonly, the PV panels of a PV-powered charging station face the sun at an ideal angle. In the Northern Hemisphere, for PV panels at a charging station, $\psi = 180°$ (alignment towards the south). In this way, as

seen in Figure 11, on a sunny day, the highest solar radiation level can be obtained around 1:00 p.m. (daylight saving time).

Figure 21 shows a model from Fujimi, a manufacturer of the Toyota Prius, at a scale of 1:24. This model is used to illustrate the impact of the vehicle's cardinal direction on the efficiency of the PV installation. The location of the standard 156 mm × 156 mm poly-Si along the roof of the vehicle is indicated in the figure. Figure 22 shows the simulated PV energy from the different rows of PV cells if the vehicle were oriented towards the south ($\psi = 180°$).

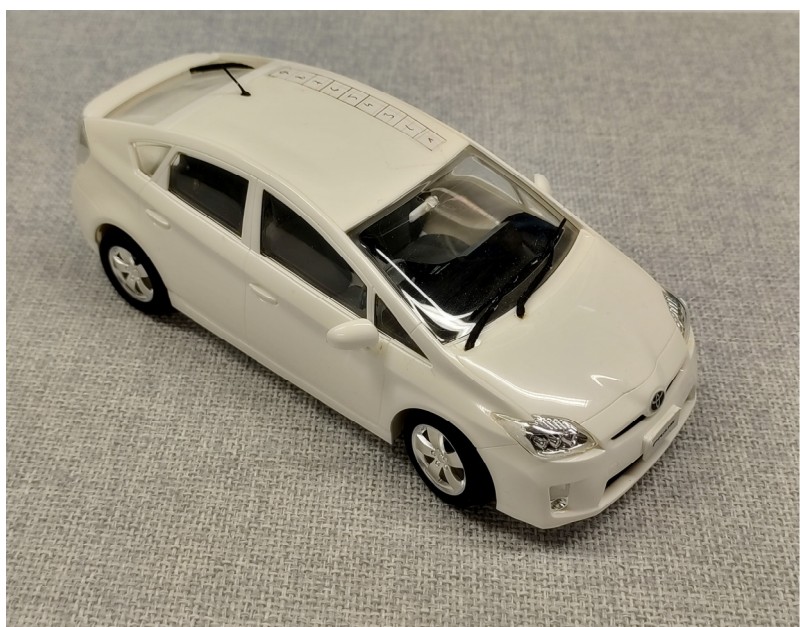

**Figure 21.** Toyota Prius model from Fujimi at a scale of 1:24.

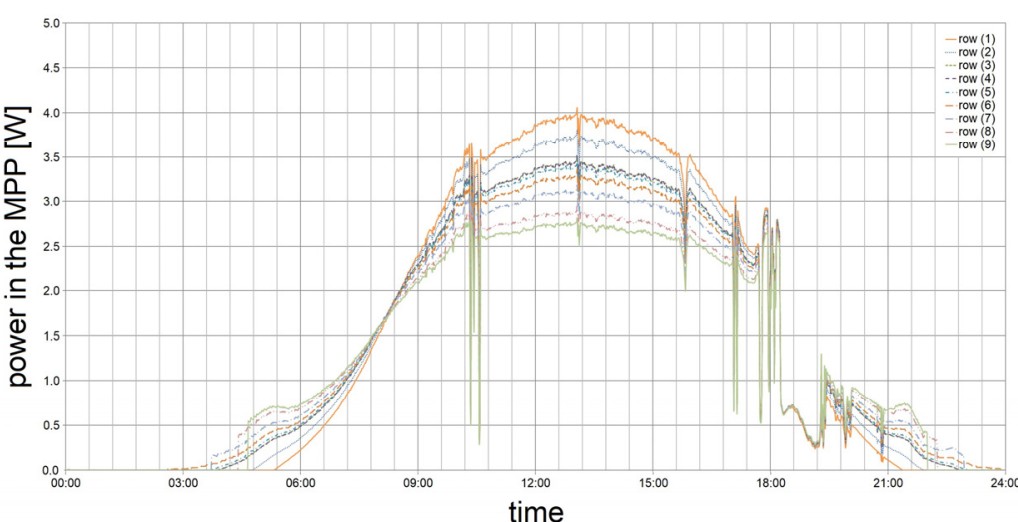

**Figure 22.** Simulated power at MPP ($P_{mpp}$) for $\psi = 180°$.

It can be seen that, for the PV cell in row 1 with $\alpha = 15°$, the highest amount of power at the MPP ($P_{mpp}$) can be obtained. For example, in the morning, if the driver orients the vehicle towards the southeast ($\psi = 135°$), as with the model in Figure 21, the output performance, or the power at the MPP of the PV installation, can be improved, as shown in Figure 23.

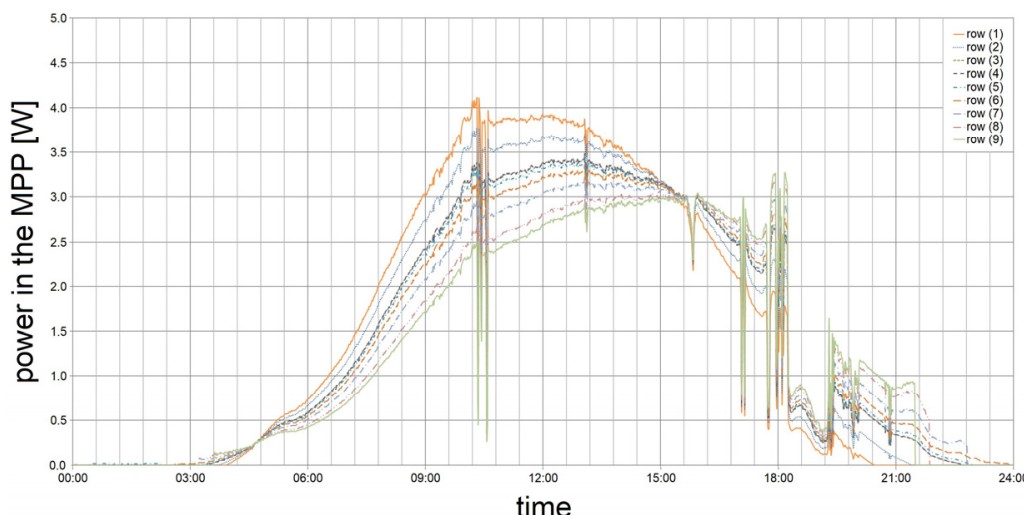

**Figure 23.** Simulated power at MPP ($P_{mpp}$) for $\psi = 135°$.

Comparing the simulated results in Figures 22 and 23, we can see, for example, that at 10:00 a.m., PV cells in row one produce about 0.5 W, or 15% more, with $\psi = 135°$ than with $\psi = 180°$. However, this increase in output power only occurs in the morning. In the afternoon, because the vehicle is oriented to the southeast and facing away from the sun, the output power will be lower.

Figure 24 shows the worst-case scenario that is possible when the vehicle is oriented towards the north. Even though the output power of PV cells in rows 6–9 is improved, the output power of PV cells in rows 1–4 will be reduced. As a result, the total output power of the PV installation is reduced. With $\alpha = 0°$, PV cells in row six are unaffected by the vehicle's cardinal orientation.

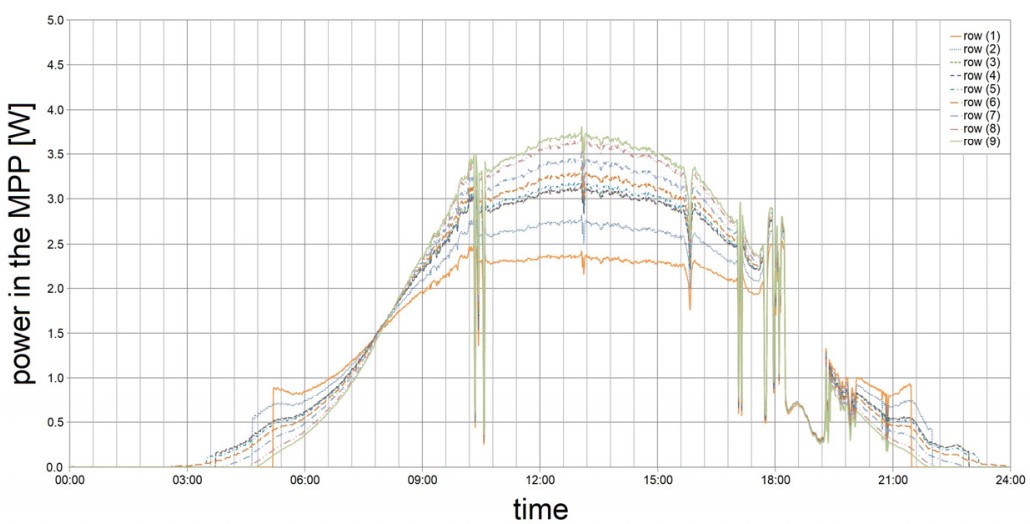

**Figure 24.** Simulated power at MPP ($P_{mpp}$) for $\psi = 0°$.

As seen on the model in Figure 21 and based on the simulation results shown in Figures 22–24, PV cells on top of the curved surface are affected by the shape of the roof and the resulting longitudinal angles of PV cells, and the vehicle's orientation relative to the sun. Theoretically speaking, the driver can improve the output power of the PV installation for a certain period of time if the car is oriented at a beneficial angle towards the sun, as shown in Figure 19.

In driving conditions, the roof can receive an uneven amount of irradiation. Even though the width of the roof is small, some parts will receive more irradiation on average

than other parts. Figure 25 illustrates the distribution of the average solar radiation level when driving from the sports hall to the botanical garden, as seen in Figure 19. As in Reference [13], each part of the roof was monitored by one sensor unit, from which the average was calculated.

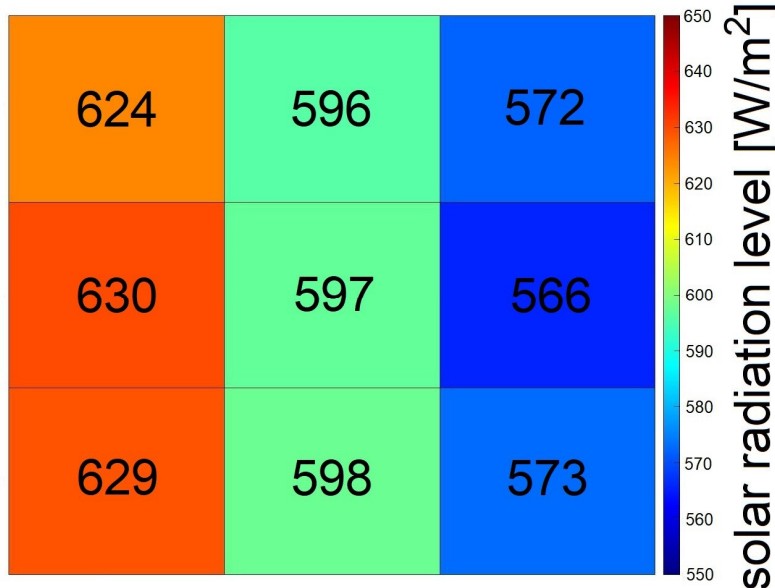

**Figure 25.** Average solar radiation levels obtained by individual sensor units.

As seen in Figure 25, due to shade from trees on the right-hand side of the street, that side will receive the least solar radiation. The middle area of the roof will receive more irradiation, but still less than the left side of the roof. These circumstances need to be taken into account when connecting PV cells together to form a PV panel. It is worth noting that the output current of a PV cell will be determined by the weakest cell in the interconnection [10].

*5.2. Discussion of the Impacts on the Degree of Efficiency of PV-Powered Vehicles*

At present, PV installations integrated on top of electric vehicles still represent a new area for PV energy systems [9]. While concepts are often discussed, in this paper, we present measurement data illustrating some of the major challenges that need to be solved in order to provide PV-powered vehicles with enough PV energy. The PV energy gained could reduce the need for energy from the power grid and, therefore, reduce the amount of load that an electric vehicle represents to the grid [3].

This paper shows environmental data obtained in Oulu, Finland. It is worth noting that, in other geographical areas, the environmental conditions may be different. For example, in urban areas, shade from the surrounding environment will occur, reducing the amount of energy obtained from the vehicle's PV installation. While PV-powered charging stations are subject to stationary conditions (i.e., parking conditions), PV-powered electric vehicles are also subject to dynamic conditions (i.e., driving conditions).

The PV energy will be lower in driving conditions than in parking conditions. As seen in Figures 19 and 20, the solar radiation level ($\lambda$) in parking conditions is approximately 680 W/m$^2$; in Figure 19, the average level is about 597 W/m$^2$; and in Figure 20, it is about 648 W/m$^2$. Various parameters, such as the driving direction, time of the day, time of the year, geographical location, etc., will affect the PV energy from the vehicle's installation.

Due to the different orientation of PV cells on top of a curved surface, the amount of energy between individual PV cells will differ. As such, it will be challenging to determine the type of connection of cells to form a PV panel. In driving conditions, the solar radiation level will change rapidly. More precisely, whereas the time window for stationary PV installations is in seconds, the time window for moving PV installations is in milliseconds [13].

As a result, even though the average solar radiation level is still notable, due to limitations with the MPPT tracker, the degree of efficiency will be reduced [13,15].

Commonly, MPPT tracking algorithms update parameters on a regular basis. For example, in voltage-based maximum power point tracking (VMPPT), the power converter is disconnected to measure the open-circuit voltage ($V_{oc}$). The voltage in the MPP ($V_{mpp}$) is estimated with the help of a multiplication factor, and then the operating voltage ($V_{op}$) is adjusted to match $V_{mpp}$. Likewise, in the perturb-and-observe (P&O) algorithm, the output voltage is measured, and, depending on the change in power, the $V_{oc}$ is either increased or decreased [32].

In driving conditions, for frequently changing environmental conditions, as seen in Figures 19 and 20 between 150 and 180 s, when measuring $P_{out}$ at 6000 samples per second, the degree of efficiency of the P&O algorithm ($\eta_{mppt}$) is about 99.67%. At 1000 samples per second, $\eta_{mppt} = 98.61\%$, and at 100 samples per second, $\eta_{mppt} = 98.23\%$. Moreover, lowering the sampling rates further results in a $\eta_{mppt}$ as low as 79.64%.

The PV cell is disconnected from the power converter to measure $P_{out}$, hence lowering $\eta_{mppt}$. This reduction has not been taking into consideration in simulations. However, in driving conditions, $V_{mpp}$ changes quickly and rapidly. For example, if $P_{out}$ would be obtained more frequently, then the PV cell is disconnected from the power converter for a too long period of time, resulting in a very low degree of efficiency. Obtaining $P_{out}$ from a reference cell is difficult, as the solar radiation level can be different from the main PV cell or panel.

The P&O algorithm can fail under rapidly changing environmental conditions [32]. As a possible solution, $V_{op}$ could be kept constant at a voltage level that favors higher solar radiation levels. Doing so would eliminate the requirement to update parameters frequently, and the power converter could be connected to the PV cell at all times. The overall system efficiency would be than at about 95% under rapidly changing environmental conditions.

PV simulation models can help to estimate potential extensions of the ERE of electric vehicles in different environmental conditions. However, as each PV cell in the installation receives a different amount of solar radiation, the cells also need to be simulated individually [33]. If PV cells are connected in series, the cell with the least irradiation will determine the output current ($I_{out}$) of the connection. If a shaded PV cell is disconnected, the degree of efficiency can be improved [10].

The sample rate of the environmental data is important. In stationary conditions, a rate of one sample every minute can be appropriate [24], while moving conditions require 1000 samples per second [13]. Environmental data can be used to analyze different environmental conditions within urban areas. Ambient data can also be used for energy-efficient routing of PV-powered vehicles in urban areas. It is worth noting that a shorter driving distance does not necessarily result in lower energy consumption [18].

## 6. Conclusions

Environmental data and computer simulations are helpful in understanding the circumstances of PV-powered electric vehicles. In parking conditions, the orientation of the vehicle relative to the sun can affect the PV energy obtained. Likewise, in driving conditions, the direction and choice of route to the destination can have a significant impact on the potential energy from the PV installation. As the available area on top of electric vehicles, such as plug-in HEVs, BEVs and CAVs, is limited, we need to develop methods and tools that will allow us to investigate the given environmental conditions for such vehicles.

As a result, PV-powered electric vehicles can be feasible, and the energy obtained from their PV installations can be significant to extend the ERE. The degree of efficiency of PV-powered electric vehicles can be optimized and maximized by taking into account the curved shape of the vehicle roof, the possible area for PV cells and the type of connection. In future work, we will continue studying the circumstances for PV-powered electric vehicles.

In particular, we will investigate potential ways to improve the degree of efficiency in driving conditions.

**Author Contributions:** Conceptualization, C.S. and T.F.; methodology, C.S. and T.F.; software, C.S.; validation, C.S.; formal analysis, C.S. and T.F.; investigation, C.S.; resources, C.S.; data curation, C.S.; writing—original draft preparation, C.S.; writing—review and editing, C.S. and T.F.; visualization, C.S.; supervision, T.F.; project administration, T.F.; funding acquisition, T.F. All authors have read and agreed to the published version of the manuscript.

**Funding:** Dr. Christian Schuss was funded by the Academy of Finland 6Genesis (6G) project (grantno. 318927). Prof. Tapio Fabritius is partially supported by Academy of Finlands FIRI funding (grantno. 320017).

**Institutional Review Board Statement:** Not applicable.

**Informed Consent Statement:** Not applicable.

**Acknowledgments:** The authors would like to express their gratitude for M. Schuss for her help and support in setting up and carrying out the measurements. We appreciate Bernd Eichberger, Harald Gall, Klaus Eberhart and Hannes Illko for their help and inputs on this research work.

**Conflicts of Interest:** The authors declare no conflict of interest.

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
