# Peer review of "Impact of Environmental Conditions on the Degree of Efficiency and Operating Range of PV-Powered Electric Vehicles"

_applsci, doi:10.3390/app12031232_

Round 1

Reviewer 1 Report

It is necessary to analyze the effect on PV-powered electric vehicle efficiency, but there are concerns about manuscript novelty and contribution.
Authors must address the following issues:

  1. What is the novelty of the manuscript? Also, please clarify the academic contribution of this manuscript.
  2. I am concerned about the generalization of the results. In particular, unlike parking lots, the consequences of driving conditions are greatly influenced by various environmental conditions. Although the author mentions that various environmental conditions, this manuscript deals with to be of very limited scope.
  3. On page 15, line 310, what are the limitations of MPPT that reduce efficiency?
  4. In section 5.2, the authors provide a discussion of the effectiveness of PV-powered electric vehicles, but It covers the general discussion. The authors must add discussions considering the findings.
  5.  English language review is required.

Author Response

First, we would like to thank the reviewer for their time and help to improve the quality of our manuscript.

  1. According to your comment, we extended the introduction section. We know highlight in a better way the knowledge, therefore our contribution to the respective research field.
  2. Thank you for your comment. It is true that environmental conditions in driving conditions differ significantly from those ones in parking conditions. While we are not able to include all parameter into our considerations, we concentrate on the most important ones. For example for MPPT algorithms, the rate of change of the solar radiation level is crucial.
  3. We added a paragraph to this section to address your comment.
  4. In order to address your comment, we extended Section 5.2 by three paragraphs.
  5. We utilised the language editing service of the journal to revise the English language of our manuscript.

Reviewer 2 Report

Paper is interesting, but it is preliminary and lots of issues are not discussed.   Some suggestions:
  • Section 3.2 is unuseful for the purposes of the paper, because they are not all used. It needs to specify which PV model is used for simulations reported in fig. 19 and successive ones.
  • Eq (8) should be supported by a figure reporting all the cited angles.
  • In section 4, it should be better that subsections 4.1 and 4.3 (parking conditions) are close; so, Subsections 4.2 and 4.4 (driving condition) will be close.
  • At line 240, word "Figure" is missing. 
  • Evaluations about the efficiency in driving conditions, in section 5.2, are not supported by data; so, I suggest to delete them or to improve the paper with other data supporting them.

Author Response

First, we would like to thank the reviewer for their time and help to improve the quality of our manuscript.

  1. Thank you for your comment. We added Section 3.2.5 in which we describe which model was used in our work.
  2. We included three new figures to our manuscript to address your comment.
  3. According to your suggestion, we rearranged the order of the subsections in Section 4.
  4. We added the missing word in that line.
  5. As you suggested, we added more data and information to Section 5.2.

Round 2

Reviewer 1 Report

My comments were properly resolved.

Reviewer 2 Report

The improved version addressed the previous criticalities.